# LANGUAGE MODELING TEACHES YOU MORE SYNTAX THAN TRANSLATION DOES

## ABSTRACT

Recent work using auxiliary prediction task classifiers to investigate the properties of LSTM representations has begun to shed light on why pretrained representations, like ELMo (Peters et al., 2018) and CoVe (McCann et al., 2017), are so beneficial for neural language understanding models. We still, though, do not yet have a clear understanding of how the choice of pretraining objective affects the type of linguistic information that models learn. With this in mind, we compare four objectives—language modeling, translation, skip-thought, and autoencoding—on their ability to induce syntactic and part-of-speech information. We make a fair comparison between the tasks by holding constant the quantity and genre of the training data, as well as the LSTM architecture. We find that representations from language models consistently perform best on our syntactic auxiliary prediction tasks, even when trained on relatively small amounts of data. These results suggest that language modeling may be the best data-rich pretraining task for transfer learning applications requiring syntactic information. We also find that the representations from randomly-initialized, frozen LSTMs perform strikingly well on our syntactic auxiliary tasks, but this effect disappears when the amount of training data for the auxiliary tasks is reduced.

## 1 INTRODUCTION

Representation learning with deep recurrent neural networks has revolutionized natural language processing and replaced many of the expert-designed, linguistic features used previously. Recently, researchers have begun to investigate the properties of learned representations by training auxiliary classifiers that use the hidden states of frozen, pretrained models to perform other tasks. These investigations have shown that when deep LSTMs (Hochreiter & Schmidhuber, 1997) are trained on tasks like translation, they learn substantial syntactic and semantic information about their input sentences, including part-of-speech (Shi et al., 2016; Belinkov et al., 2017a;b; Blevins et al., 2018).

These intriguing findings lead us to ask the following questions:

1. How does the training task affect how well models learn syntactic properties? Which tasks are better at inducing these properties?
2. How does the amount of data the model is trained on affect these results? When does training on more data help?

We investigate these questions by holding the data source and model architecture constant, while varying both the training task and the amount of training data. Specifically, we examine models trained on English-German (En-De) translation, language modeling, skip-thought (Kiros et al., 2015), and autoencoding, and also compare to an untrained LSTM model as a baseline. We control for the data domain by exclusively training on datasets from the 2016 Conference on Machine Translation (WMT; Bojar et al., 2016). We train models on all tasks using the parallel En-De corpus, which allows us to make fair comparisons across tasks. We also train models on a subset of the this corpus to examine the effect of training data volume on learned representations. Additionally, we augment the parallel dataset with a large monolingual corpus from WMT to examine how the performance of the unsupervised tasks (all but translation) scale with more data.

Throughout our work, we focus on the syntactic evaluation tasks of part-of-speech (POS) tagging and Combinatorial Categorical Grammar (CCG) supertagging. Supertagging is considered a build-

ing block for parsing as these tags constrain the ways in which words can compose, largely determining the parse of the sentence. CCG supertagging thus allows us to measure the degree to which models learn syntactic structure above the word. We focus our analysis on representations learned by language models and by the *encoders* of sequence-to-sequence models, as translation encoders have been found to learn richer representations of POS and morphological information than translation decoders (Belinkov et al., 2017a).

We find that for POS and CCG tagging, bidirectional language models (BiLMs)—created by separately training forward and backward language models, and concatenating their hidden states—outperform models trained on all other tasks. Even BiLMs trained on relatively small amounts of data (1 million sentences) outperform translation and skip-thought models trained on larger datasets (5 million and 63 million sentences respectively).

Our inclusion of an untrained LSTM baseline allows us to study the effect of *training* on hidden state representations of LSTMs. We find, surprisingly, that when we use all of the available labeled tag data to train our auxiliary task classifiers, our best trained models (BiLMs) only outperform the randomly initialized, untrained LSTMs by a few percentage points. When we reduce the amount of classifier training data though, the performance of the randomly initialized LSTM model drops far below those of trained models. We hypothesize that this occurs because training the classifiers on large amounts of auxiliary task data allows them to memorize configurations of words seen in the training set and their associated tags. We test this hypothesis by training classifiers to predict the identity of neighboring words from a given hidden state, and find that randomly initialized models *outperform* all trained models on this task. Our findings demonstrate that our best trained models do well on the tagging tasks because they are truly learning representations that conform to our notions of POS and CCG tagging, and not simply because the classifiers we train are able to recover neighboring word identity information.

## 2 RELATED WORK

**Evaluating Learned Representations**     Adi et al. (2016) introduce the idea of examining sentence vector representations by training auxiliary classifiers to take sentence encodings and predict attributes like word order. Belinkov et al. (2017a) build on this work by examining the hidden states of LSTMs trained on translation and find that they learn substantial POS and morphological information without direct supervision for these linguistic properties. Beyond translation, Blevins et al. (2018) find that deep LSTMs learn hierarchical syntax when trained on a variety of tasks—including semantic role labeling, language modeling, and dependency parsing. However, the models examined by Blevins et al. (2018) were also trained on different datasets, so it's unclear if the differences in syntactic task performance are due to the training objectives or simply differences in the training data. By controlling for model size and the quantity and genre of the training data, we we are able to make direct comparisons between tasks on their ability to induce syntactic information.

**Transfer Learning of Representations**     Much of the work on sentence-level pretraining has focused on sentence-to-vector models and evaluating learned representations on how well they can be used to perform sentence-level classification tasks. A prominent early success in this area with unlabeled data is skip-thought (Kiros et al., 2015), the technique of training a sequence-to-sequence model to predict the sentence preceding and following each sentence in a running text. InferSent (Conneau et al., 2017)—the technique of pretraining encoders on natural language inference data—yields strikingly better performance when such labeled data is available.

Work in transfer learning of representations has recently moved beyond strict sentence-to-vector mappings. Newer models that incorporate LSTMs or Transformer networks pretrained on data-rich tasks, like translation and language modeling, have achieved state-of-the-art results on many tasks—including semantic role labeling, natural language inference, and coreference resolution (Peters et al., 2018; McCann et al., 2017; Howard & Ruder, 2018; Radford et al., 2018). Although comparisons have previously been made between translation and language modeling as pretraining tasks (Peters et al., 2018; Wang et al., 2018), we investigate this issue more thoroughly by controlling for the quantity and content of the training data.

| Task | Layer Size | Attn. | 1 Million | 5 Million | 15 Million | 63 Million |
|---|---|---|---|---|---|---|
| Translation | 2×500D | Y | 13.2 (17.6 BLEU) | 9.1 (21.4 BLEU) | – | – |
| Translation | 2×500D | N | 25.2 (6.8 BLEU) | 13.0 (12.3 BLEU) | – | – |
| LM Forward | 1×500D | – | 104.8 | 81.2 | 82.3 | 76.9 |
| LM Backward | 1×500D | – | 103.2 | 80.8 | 81.1 | 77.3 |
| LM Forward | 1×1000D | – | 103.8 | 73.6 | 69.2 | 66.5 |
| Skip-Thought | 2×500D | Y | 99.0 | 69.2 | 68.7 | 67.9 |
| Skip-Thought | 2×500D | N | 104.1 | 72.0 | 68.1 | 66.7 |
| Autoencoder | 2×500D | Y | 1.0 | 1.0 | 1.0 | 1.0 |
| Autoencoder | 2×500D | N | 1.0 | 1.1 | 1.2 | 1.1 |

Table 1: Perplexity of trained models by number of training sentences. All but the language models are 1000D BiLSTMs (500D per direction). The 500D forward and backward language models are combined into a single bidirectional language model for analysis experiments.

**Training Dataset Size** The performance of neural models depends immensely on the amount of training data used. Koehn & Knowles (2017) find that when training machine translation models on corpora with fewer than 15 million words (English side), statistical machine translation approaches outperform neural ones. Similarly, Hestness et al. (2017) study the affect of training data volume on performance for several tasks—including translation and image classification. They find that for small amounts of data, neural models perform about as well as chance. After a certain threshold, model performance improves logarithmically with the amount of training data, but this eventually plateaus. With this in mind, we also vary the amount of training data to investigate the relationship between performance and data volume for each task.

**Randomly Initialized Models** Conneau et al. (2018) use randomly initialized LSTMs as a baseline when studying sentence-to-vector embedding models. They find that untrained models outperform many trained models on several auxiliary tasks, including predicting word content. Similarly in vision, untrained convolutional networks have been shown to capture many low-level image statistics and can be used for image denoising (Ulyanov et al., 2017). Our method of training auxiliary classifiers on randomly initialized RNNs builds on the tradition of reservoir computing, in which randomly initialized networks or "reservoirs" are fixed and only "read-out" classifier networks are trained (Lukoševičius & Jaeger, 2009). Echo state networks—reservoir computing with recurrent models—have been used for tasks like speech recognition, language modeling, and time series prediction (Verstraeten et al., 2006; Tong et al., 2007; Sun et al., 2017).

# 3 METHODS

## 3.1 MAIN TRAINING DATA

We use the parallel English-German (En-De) dataset from the 2016 ACL Conference on Machine Translation (WMT) shared task on news translation (Bojar et al., 2016). This dataset contains 5 million ordered sentence translation pairs. We also use the 2015 English monolingual news discussion dataset from the same WMT shared task, which contains approximately 58 million ordered sentences. To examine how the volume of training data affects learned representations, we use four corpus sizes: 1, 5, 15, and 63 million sentences (translation is only trained on the smaller two sizes). We create the 1 million sentence corpora from the 5 million sentence dataset by sampling (i) sentence pairs for translation, (ii) English sentences for autoencoders, and (iii) ordered English sentence pairs for skip-thought and language models[1]. Similarly, we create the 15 million sentence corpora for the unsupervised tasks by sampling sentences from the entire corpus of 63 million sentences. We use word-level representations throughout and use the Moses package (Koehn et al., 2007) to tokenize and truecase our data. Finally, we limit both the English and German vocabularies to the 50k most frequent tokens in the training set.

---

[1]Note, in training we initialize the language model LSTM hidden states with the final state after reading the previous sentence.

Soon she was running the office
RB PRP VBD VBG DT NN

(a) POS tags

Soon she was running the office
S/S NP (S\NP)/NP NP NP/N N
>
(S\NP) / NP NP
>
S\NP
<
S
>
S

(b) A CCG parse, with supertags shown immediately below the words.

Figure 1: An annotated PTB example sentence.

## 3.2 Model Architecture and Training

We train all our models using OpenNMT-py (Klein et al., 2017) and use the default options for model sizes, hyperparameters, and training procedure—except we increase the size of the LSTMs, make the encoders bidirectional, and use validation-based learning rate decay instead of a fixed schedule. Specifically, all our models (except language models) are 1000D, two-layer encoder-decoder LSTMs with bidirectional encoders (500D per direction) and 500D embeddings. We train models both with and without attention (Bahdanau et al., 2015). For language models, we train a 1000D forward language model and a bidirectional language model—two 500D language models (forward and backward) trained separately, whose hidden states are concatenated. All models, including our untrained baseline, are initialized from a uniform distribution $(-0.1, 0.1)$, the default in OpenNMT.

We use the same training procedure for all our models. We evaluate on the validation set every epoch when training on the 1 and 5 million sentence datasets, and evaluate approximately every 5 million sentences when training on the larger datasets. We use SGD with an initial learning rate of 1. Whenever a model's validation loss increases relative to the previous evaluation, we halve the learning rate and stop training when the learning rate reaches $0.5^{15}$. For each training task and dataset size, we select the model with the lowest validation perplexity to perform auxiliary task evaluations on. We report model performance in terms of perplexity and BLEU (Papineni et al., 2002) in Table 1. For translation we use beam search ($B = 5$) when decoding.

## 3.3 Classifier Data and Architecture

**POS and CCG** For Part-of-Speech (POS) tagging evaluation, we use the Wall Street Journal (WSJ) portion of the Penn Treebank (PTB; Marcus et al., 1993) We follow the standard WSJ split (train 2-21; dev 22; test 23). The dataset contains approximately 50k sentences and 45 tag types.

For CCG supertagging, we use CCG Bank (Hockenmaier & Steedman, 2007), which is based on PTB WSJ. CCG supertagging provides fine-grained information about the role of each word in its larger syntactic context and is considered *almost* parsing, since sequences of tags map sentences to small subsets of possible parses. The entire dataset contains approximately 50k sentences and 1327 tag types. We display POS and CCG tags for an example sentence in Figure 1.

To study the impact of auxiliary task training data volume, for both datasets we create smaller classifier training sets by sampling 10% and 1% of the sentences. We truecase both datasets using the same truecase model trained on WMT and restrict the vocabularies to the 50k tokens used in pretraining our LSTM models. In addition to the untrained LSTM baseline, we also compare to the word-conditional most frequent class (WC-MFC)—the most frequently assigned tag class for each distinct word in the training set. For this baseline we restrict the vocabulary to that of our LSTM models and map all out-of-vocabulary words to a single UNK token. Note that while PTB and WMT are both drawn from news text, there is slight genre mismatch.

**Word Identity** For this task, the classifier takes a single LSTM hidden state as input and predicts the identity of the word at a different time step. For example, for the sentence "I love NLP" and a time step shift of -2, we would train the classifier to take the hidden state for "NLP" and predict the word "I". We use the WSJ dataset for this task. Following Conneau et al. (2018), we take all words that occur between 100 and 1000 times (about 1000 words total) as the possible targets for neighboring word prediction.

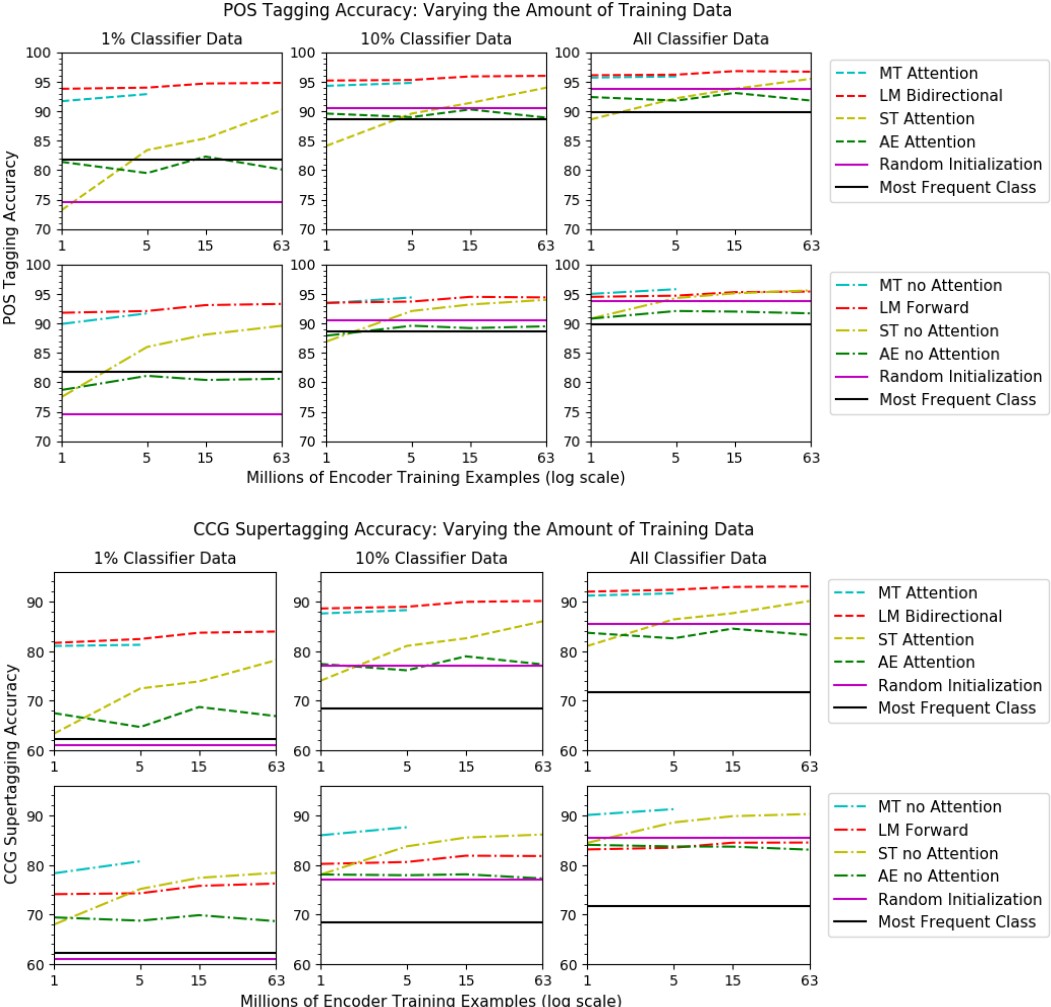

Figure 2: POS and CCG tagging accuracies for different amounts of LSTM encoder and classifier training data. We show results for the best performing layer of each model. Note, BiLMs are displayed with the attention models and forward LMs are displayed with the models without attention.

**Classifier Training Procedure**   We train multi-layer perceptron (MLP) classifiers that take an LSTM hidden state (from one time step and one layer) and output a distribution over the possible labels (tags or word identities). The MLPs we train have a single 1000D hidden layer with a ReLU activation. For classifier training, we use the same training and learning rate decay procedure used for pretraining the LSTM encoders.

## 4   COMPARING PRETRAINING TASKS

In this section we discuss the main POS and CCG tagging results displayed in Figure 2. Overall, POS and CCG tagging accuracies tend to increase with the amount of data the LSTM encoders are trained on, but the marginal improvement decreases as the amount of training data increases.

**Language Modeling and Translation**   For all pretraining dataset sizes, bidirectional language model (BiLM) and translation encoder representations perform best on both POS and CCG tagging. Translation encoders, however, slightly underperform BiLMs, even when both models are trained on the same amount of data. In fact, even BiLMs trained on the smallest amount of data (1 million sentences) outperform models trained on all other tasks and dataset sizes (up to 5 million sentences for translation, and 63 million sentences for skip-thought and autoencoding). Especially since BiLMs

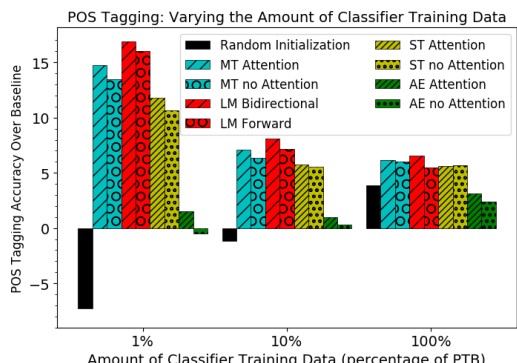 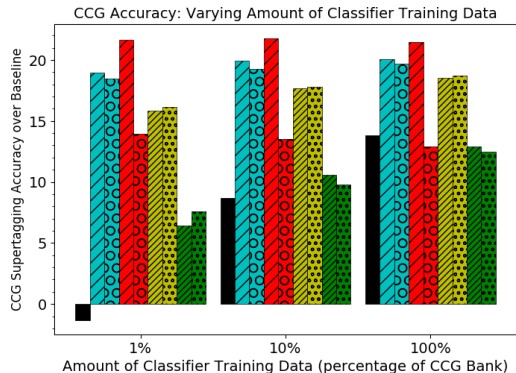

(a) WC-MFC baselines for different amounts of PTB training data: 1% PTB: 81.8%; 10% PTB: 88.6%; 100% PTB: 89.9%.

(b) WC-MFC baselines for different amounts of CCG training data: 1% CCG: 62.3%; 10% CCG: 68.3%; 100% CCG: 71.6%.

Figure 3: POS and CCG tagging accuracies for different amounts of classifier training data in terms of percentage points over the word-conditional most frequent class (WC-MFC) baseline. We show results for the best performing layer and model for each task.

do not require aligned data to train, the superior performance of BiLM representations on these tasks suggests that BiLMs (like ELMo; Peters et al., 2018) are better than translation encoders (like CoVe; McCann et al., 2017) for transfer learning of syntactic information. One reason BiLMs perform relatively well on these syntactic tasks could be that in contrast to the encoders for the other tasks, LM encoders have a per-token loss. Note also that since our evaluation tasks also predict a single label for each token, this could be one reason that BiLMs perform so well on these tasks in particular.

For all amounts of training data, the BiLMs significantly outperform the 1000D forward-only language models. The gap in performance between bidirectional and forward language models is greater for CCG supertagging than for POS tagging. When using all available auxiliary training data, there is a 2 and 8 percentage point performance gap in POS and CCG tagging respectively. This difference in relative performance suggests that bidirectional context information is more important for identifying syntactic structure than for identifying part of speech.

Figure 2 illustrates how the best performing BiLMs and translation models tend to be more robust to decreases in classifier data than models trained on other tasks. Also, when training on less auxiliary task data, POS tagging performance tends to drop less than CCG supertagging performance. For the best model (BiLM trained on 63 million sentences), when using 1% rather than all of the auxiliary task training data, CCG accuracy drops 9 percentage points, while POS accuracy only drops 2 points. Further examinations of the effect of classifier data volume are displayed in Figure 3.

**Skip-Thought** Although skip-thought encoders consistently underperform both BiLMs and translation encoders in all data regimes we examine, skip-thought models improve the most when increasing the amount of pretraining data, and are the only models whose performance does not seem to have plateaued by 63 million training sentences. Since we train our language models on ordered sentences, as we do for skip-thought, our language models can be interpreted as a regularized versions of skip-thought, in which the weights of the encoder and decoder are shared. The increased model capacity of skip-thought, compared to language models, could explain the difference in learned representation quality—especially when these models are trained on smaller amounts of data.

**Random Initialization** For our randomly initialized, untrained LSTM encoders, we use the default weight initialization technique in OpenNMT-py, a uniform distribution between -0.1 and 0.1; the only change we make is to set all biases to zero. We find that this baseline performs quite well when using all auxiliary data, and is only 3 and 8 percentage points behind the BiLM on POS and CCG tagging, respectively. We find that decreasing the amount of classifier data leads to a significantly greater drop in the untrained encoder performance compared to trained models. In the 1% classifier data regime, the performance of untrained encoders on both tasks drops below that of all trained models and below even the word-conditional most-frequent class baseline.

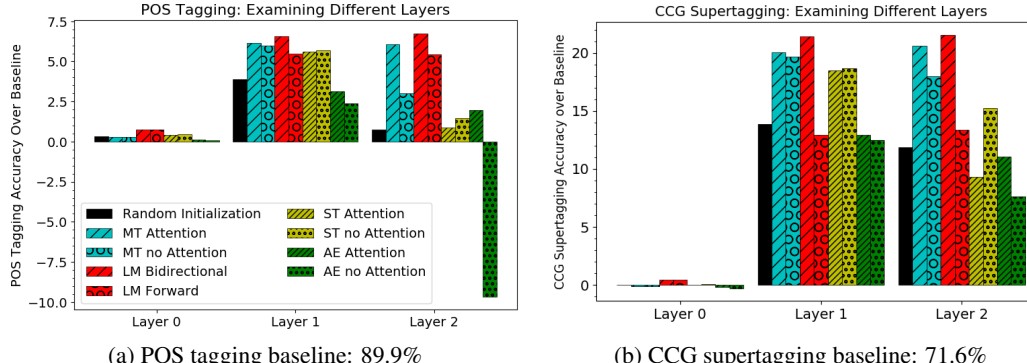

(a) POS tagging baseline: 89.9%  (b) CCG supertagging baseline: 71.6%

Figure 4: POS and CCG tagging accuracies in terms of percentage points over the word-conditional most frequent class baseline. We display results for the best performing models for each task.

We hypothesize that the randomly initialized baseline is able to perform well on tagging tasks with large amounts of auxiliary task training data, because the classifier can learn the identity of neighboring words from a given time step's hidden state, and simply memorize word configurations and their associated tags from the training data. We test this hypothesis directly in Section 6 and find that untrained LSTM representations are in fact better at capturing neighboring word *identity* information than any trained model.

**Autoencoder**   Models trained on autoencoding are the only ones that do not consistently improve with the amount of training data, which is unsurprising as unregularized autoencoders are prone to learning identity mappings (Vincent et al., 2008). When training on 10% and 1% of the auxiliary task data, autoencoders outperform randomly initialized encoders and match the word-conditional most frequent class baseline. When training on *all* the auxiliary data though, untrained encoders outperform autoencoders. These results suggest that autoencoders learn some useful structure that is useful in the low auxiliary data regime. However, the representations autoencoders learn do not capture syntactically rich features, since random encoders outperform them in the high auxiliary data regime. This conclusion is further supported by the extremely poor performance of the second layer of an autoencoder without attention on POS tagging (almost 10 percentage points below the most frequent class baseline), as seen in Figure 4a.

## 5   COMPARING LAYERS

**Embeddings (Layer 0)**   We find that randomly initialized embeddings consistently perform as well as the word-conditional most frequent class baseline on POS and CCG tagging, which serves as an upper bound on performance for the embedding layer. As these embeddings are untrained, the auxiliary classifiers are learning to memorize and classify the random vectors. When using all the auxiliary classifier data, there is no significant difference in the performance of trained and untrained embeddings on the tagging tasks. Only for smaller amounts of classifier data do trained embeddings consistently outperform randomly initialized ones.

**Upper Layers**   Belinkov et al. (2017a) find that, for translation models, the first layer consistently outperforms the second on POS tagging. We find that this pattern holds for all our models, except in BiLMs, for which the first and second layers perform equivalently. The pattern holds even for untrained models, suggesting that POS information is stored on the lower layer, not necessarily because the training task encourages this, but because of properties of the deep LSTM architecture.

We also find that for CCG supertagging, the first layer also outperforms the second layer on untrained models. For the trained models though, the second layer performs better than the first in some cases. Which layer performs best appears to be independent of absolute performance on the supertagging task. Our layer analysis results are displayed in Figure 4.

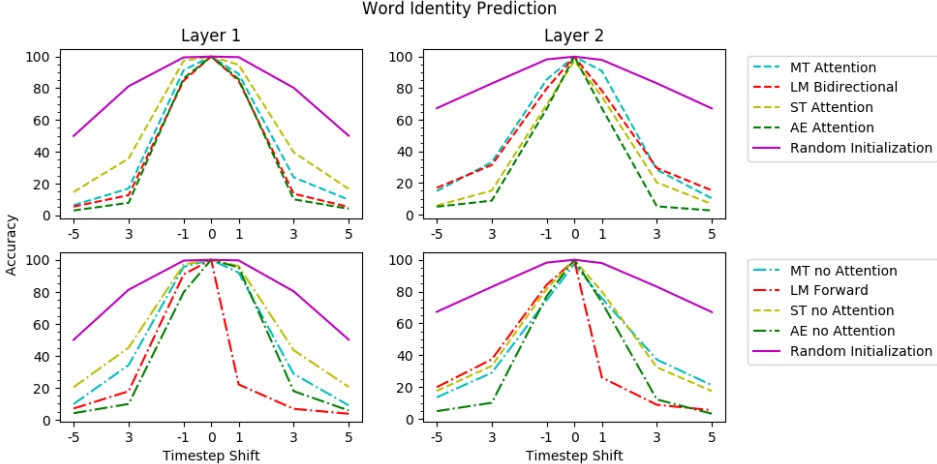

Figure 5: Performance of classifiers trained to predict the identity of the word a fixed number of timesteps away. Note, the forward LM has asymmetrical access to this information in its input.

## 6 WORD IDENTITY PREDICTION

Our results on word identity prediction are summarized in Figure 5 and given in more detail in Appendix A. While trained encoders outperform untrained ones on both POS and CCG tagging, we find that all trained LSTMs *underperform* untrained ones on word identity prediction. This finding confirms that trained encoders genuinely capture substantial syntactic features, beyond mere word identity, that the auxiliary classifiers can use.

We find that for both trained and untrained models, the first layer outperforms the second layer when predicting the identity of the *immediate* neighbors of a word. However, the second layer tends to outperform the first at predicting the identity of more distant neighboring words. This effect is especially apparent for the randomly initialized encoders. Our finding suggests that, as is the case for convolutional neural networks, depth in recurrent neural networks has the effect of increasing the receptive field and allows the upper layers to have representations that capture a larger context. These results reflect the findings of Blevins et al. (2018) that for trained models, upper levels of LSTMs encode more abstract syntactic information, since more abstract information generally requires larger context information.

## 7 CONCLUSION

By controlling for the genre and quantity of the training data, we make fair comparisons between several data-rich training tasks in their ability to induce syntactic information. We find that bidirectional language models (BiLMs) do better than translation and skip-thought encoders at extracting useful features for POS tagging and CCG supertagging. Moreover, this improvement holds even when the BiLMs are trained on substantially *less* data than competing models. Our results suggest that for transfer learning, BiLMs like ELMo (Peters et al., 2018) capture more useful features than translation encoders—and that this holds even on genres for which data is not abundant.

We also find that randomly initialized encoders extract usable features for POS and CCG tagging—at least when the auxiliary POS and CCG classifiers are themselves trained on reasonably large amounts of data. The performance of untrained models drops sharply relative to trained ones when using smaller amounts of the classifier data. We investigate further and find that untrained models outperform trained ones on the task of neighboring word identity prediction, which confirms that trained encoders do not perform well on tagging tasks because the classifiers are simply memorizing word identity information. We also find that both trained and untrained LSTMs store more *local* neighboring word identity information in lower layers and more *distant* word identity information in upper layers, which suggests that depth in LSTMs allow them to capture larger context information.

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

# A    RANDOMLY INITIALIZED ENCODERS

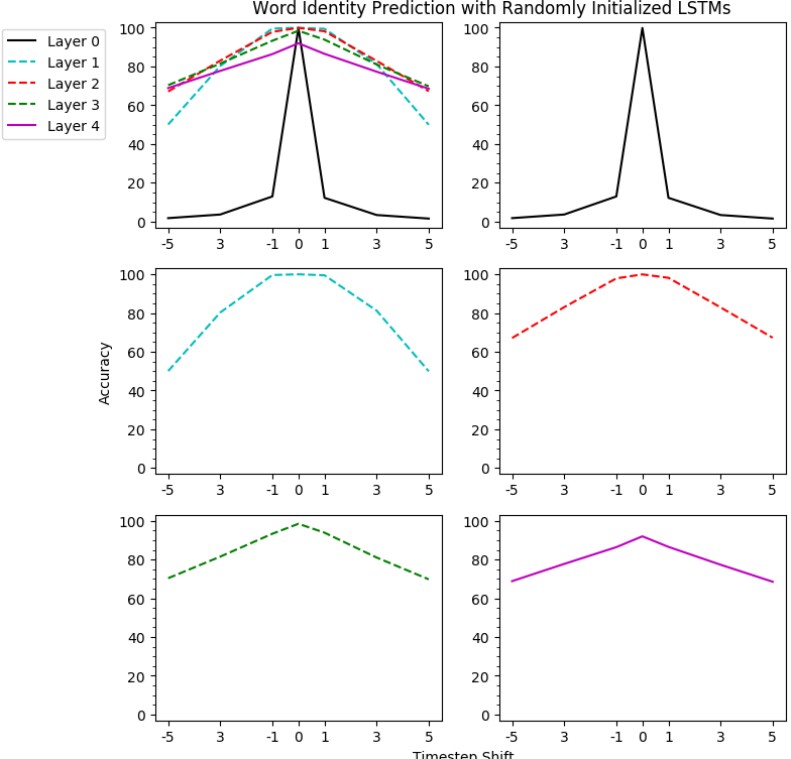

Figure 6: Here we display results for the word identity prediction task with randomly initialized LSTM encoders with up to 4 layers. Lower layers have a more peaked shape and upper layers a more flat shape, meaning that the lower layers encode relatively more *nearby* neighboring word information, while upper layers encode relatively more *distant* neighboring word information.

## B  POS AND CCG EVALUATION FULL RESULTS

### B.1  TRAINING CLASSIFIERS ON ALL DATA

| Training task | Data | Attention | POS L2 | POS L1 | POS L0 | CCG L2 | CCG L1 | CCG L0 |
|---|---|---|---|---|---|---|---|---|
| Random Init 1 | None | N/A | 90.5 | 93.7 | 90.2 | 83.5 | 85.4 | 71.6 |
| Random Init 2 | None | N/A | 90.3 | 93.8 | 90.1 | 83.3 | 85.3 | 71.5 |
| Translation | 1M | Yes | 95.6 | 95.7 | 90.0 | 91.4 | 91.2 | 71.5 |
| Translation | 1M | No | 92.5 | 95.0 | 90.0 | 88.2 | 90.1 | 71.3 |
| LM (Bidir) | 1M | No | 96.4 | 96.1 | 90.2 | 92.5 | 92.0 | 71.6 |
| LM (Forward) | 1M | No | 94.3 | 94.5 | 90.1 | 83.5 | 83.1 | 71.5 |
| Skip-thought | 1M | Yes | 44.3 | 88.6 | 89.9 | 45.3 | 81.0 | 71.1 |
| Skip-thought | 1M | No | 78.1 | 90.8 | 89.9 | 74.5 | 84.4 | 71.1 |
| Autoencoder | 1M | Yes | 80.8 | 92.4 | 89.6 | 73.6 | 83.7 | 71.2 |
| Autoencoder | 1M | No | 79.8 | 90.8 | 89.9 | 79.2 | 84.0 | 71.1 |
| Translation | 5M | Yes | 96.0 | 95.9 | 90.2 | 92.2 | 91.6 | 71.5 |
| Translation | 5M | No | 92.9 | 95.8 | 90.2 | 89.6 | 91.2 | 71.5 |
| LM (Bidir) | 5M | No | 96.6 | 96.2 | 90.3 | 92.6 | 92.4 | 71.6 |
| LM (Forward) | 5M | No | 94.6 | 94.7 | 90.2 | 84.0 | 83.5 | 71.5 |
| Skip-thought | 5M | Yes | 76.4 | 92.2 | 90.0 | 68.4 | 86.4 | 71.1 |
| Skip-thought | 5M | No | 86.1 | 94.3 | 90.0 | 81.2 | 88.6 | 71.2 |
| Autoencoder | 5M | Yes | 88.1 | 91.8 | 89.6 | 76.5 | 82.5 | 70.8 |
| Autoencoder | 5M | No | 70.7 | 92.1 | 89.8 | 72.7 | 83.7 | 71.0 |
| LM (Bidir) | 15M | No | **97.0** | 96.8 | 90.6 | **93.1** | 92.9 | 72.0 |
| LM (Forward) | 15M | No | 95.3 | 95.3 | 90.6 | 84.9 | 84.5 | 72.0 |
| Skip-thought | 15M | Yes | 82.3 | 93.8 | 90.2 | 70.4 | 87.6 | 71.6 |
| Skip-thought | 15M | No | 90.1 | 95.1 | 90.3 | 85.8 | 89.8 | 71.5 |
| Autoencoder | 15M | Yes | 91.9 | 93.1 | 90.1 | 82.6 | 84.5 | 71.4 |
| Autoencoder | 15M | No | 71.6 | 92.0 | 89.8 | 71.0 | 83.7 | 71.2 |
| LM (Bidir) | 63M | No | 96.9 | 96.7 | 90.6 | **93.1** | 93.0 | 72.0 |
| LM (Forward) | 63M | No | 95.3 | 95.4 | 90.6 | 84.9 | 84.5 | 72.0 |
| Skip-thought | 63M | Yes | 90.6 | 95.5 | 90.3 | 80.9 | 90.1 | 71.6 |
| Skip-thought | 63M | No | 91.6 | 95.6 | 90.3 | 86.8 | 90.3 | 71.6 |
| Autoencoder | 63M | Yes | 89.4 | 91.8 | 89.6 | 78.4 | 83.2 | 71.2 |
| Autoencoder | 63M | No | 70.2 | 91.7 | 89.9 | 70.5 | 83.1 | 71.3 |

Table 2: Here we display results for training on all of auxiliary task data. Word-conditional most frequent class baselines for this amount of training data are 89.9% for POS tagging and 71.6% for CCG supertagging. For each task, we underline the best performance for each training dataset size and bold the best overall performance.

## B.2 Training Classifiers on 10% of Data

| Training task | Data | Attention | POS L2 | POS L1 | POS L0 | CCG L2 | CCG L1 | CCG L0 |
|---|---|---|---|---|---|---|---|---|
| Random Init 1 | None | N/A | 85.0 | 90.5 | 88.3 | 71.8 | 77.0 | 68.3 |
| Random Init 2 | None | N/A | 84.9 | 90.6 | 88.3 | 72.7 | 77.0 | 68.3 |
| Translation | 1M | Yes | 93.4 | 94.3 | 89.1 | 88.4 | 87.6 | 69.5 |
| Translation | 1M | No | 89.9 | 93.4 | 89.0 | 82.9 | 86.0 | 69.5 |
| LM (Bidir) | 1M | No | 95.5 | 95.2 | 89.7 | 89.4 | 88.6 | 70.1 |
| LM Forward | 1M | No | 93.2 | 93.5 | 89.5 | 80.8 | 80.2 | 69.9 |
| Skip-thought | 1M | Yes | 34.3 | 84.1 | 88.2 | 36.7 | 74.0 | 68.3 |
| Skip-thought | 1M | No | 71.3 | 86.9 | 88.2 | 64.9 | 78.0 | 68.1 |
| Autoencoder | 1M | Yes | 77.9 | 89.6 | 87.7 | 71.5 | 77.4 | 68.3 |
| Autoencoder | 1M | No | 71.2 | 87.9 | 88.6 | 71.8 | 78.1 | 68.8 |
| Translation | 5M | Yes | 94.1 | 94.8 | 89.5 | 88.9 | 88.2 | 69.8 |
| Translation | 5M | No | 89.2 | 94.4 | 89.5 | 85.4 | 87.6 | 69.9 |
| LM (Bidir) | 5M | No | 95.7 | 95.3 | 89.8 | 89.6 | 88.9 | 70.2 |
| LM Forward | 5M | No | 93.3 | 93.7 | 89.7 | 81.4 | 80.6 | 70.1 |
| Skip-thought | 5M | Yes | 66.8 | 89.6 | 88.7 | 60.8 | 81.0 | 68.7 |
| Skip-thought | 5M | No | 81.2 | 92.1 | 88.7 | 73.4 | 83.7 | 68.7 |
| Autoencoder | 5M | Yes | 84.9 | 89.0 | 87.6 | 71.8 | 76.1 | 67.9 |
| Autoencoder | 5M | No | 65.6 | 89.6 | 88.4 | 65.8 | 77.9 | 68.3 |
| LM (Bidir) | 15M | No | **96.1** | 95.9 | 90.2 | 89.7 | 89.9 | 70.6 |
| LM Forward | 15M | No | 94.1 | 94.5 | 90.1 | 82.1 | 81.8 | 70.6 |
| Skip-thought | 15M | Yes | 72.8 | 91.4 | 89.0 | 63.2 | 82.6 | 68.9 |
| Skip-thought | 15M | No | 84.6 | 93.2 | 89.0 | 79.8 | 85.5 | 69.1 |
| Autoencoder | 15M | Yes | 88.3 | 90.3 | 88.4 | 76.6 | 78.9 | 68.7 |
| Autoencoder | 15M | No | 68.5 | 89.2 | 88.3 | 68.6 | 78.1 | 68.6 |
| LM (Bidir) | 63M | No | **96.1** | 96.0 | 90.2 | 90.0 | **90.1** | 70.7 |
| LM Forward | 63M | No | 94.3 | 94.4 | 90.2 | 82.3 | 81.8 | 70.6 |
| Skip-thought | 63M | Yes | 85.0 | 94.0 | 89.2 | 73.9 | 86.0 | 69.4 |
| Skip-thought | 63M | No | 88.0 | 94.0 | 89.3 | 81.6 | 86.1 | 69.3 |
| Autoencoder | 63M | Yes | 82.8 | 88.9 | 87.4 | 72.7 | 77.3 | 68.4 |
| Autoencoder | 63M | No | 67.2 | 89.5 | 88.5 | 66.1 | 77.2 | 68.5 |

Table 3: Here we display results for training on 10% of auxiliary task data. Word-conditional most frequent class baselines for this amount of training data are 88.6% for POS tagging and 68.3% for CCG supertagging. For each task, we underline the best performance for each training dataset size and bold the best overall performance.

## B.3 Training Classifiers on 1% of Data

| Training task | Data | Attn. | POS L2 | POS L1 | POS L0 | CCG L2 | CCG L1 | CCG L0 |
|---|---|---|---|---|---|---|---|---|
| Random Init 1 | None | N/A | 68.7 | 74.5 | 79.1 | 54.4 | 60.9 | 59.3 |
| Random Init 2 | None | N/A | 68.8 | 74.5 | 79.5 | 55.5 | 62.0 | 58.8 |
| Translation | 1M | Yes | 90.8 | 91.7 | 87.2 | 79.1 | 81.0 | 65.4 |
| Translation | 1M | No | 82.5 | 89.9 | 86.9 | 69.0 | 78.3 | 65.0 |
| LM (Bidir) | 1M | No | 93.5 | 93.8 | 89.0 | 82.8 | 81.6 | 67.1 |
| LM Forward | 1M | No | 90.8 | 91.8 | 88.5 | 74.3 | 74.1 | 66.5 |
| Skip-thought | 1M | Yes | 27.2 | 73.2 | 81.4 | 28.7 | 63.3 | 60.7 |
| Skip-thought | 1M | No | 57.8 | 77.5 | 81.3 | 47.4 | 67.9 | 61.0 |
| Autoencoder | 1M | Yes | 71.2 | 81.4 | 81.8 | 59.0 | 67.4 | 61.9 |
| Autoencoder | 1M | No | 62.2 | 78.7 | 84.2 | 60.2 | 69.4 | 63.5 |
| Translation | 5M | Yes | 92.1 | 92.9 | 88.2 | 77.3 | 81.2 | 65.7 |
| Translation | 5M | No | 82.7 | 91.7 | 88.0 | 73.5 | 80.7 | 65.9 |
| LM (Bidir) | 5M | No | 93.7 | 94.0 | 89.1 | 83.0 | 82.4 | 67.1 |
| LM Forward | 5M | No | 90.7 | 92.1 | 88.8 | 74.3 | 74.3 | 66.7 |
| Skip-thought | 5M | Yes | 55.3 | 83.4 | 84.8 | 44.5 | 72.4 | 63.0 |
| Skip-thought | 5M | No | 69.6 | 86.0 | 84.4 | 53.5 | 75.1 | 62.7 |
| Autoencoder | 5M | Yes | 67.6 | 79.5 | 80.8 | 58.8 | 64.6 | 61.0 |
| Autoencoder | 5M | No | 60.7 | 81.1 | 82.6 | 56.0 | 68.7 | 61.8 |
| LM (Bidir) | 15M | No | 94.4 | 94.7 | 89.6 | 82.8 | 83.7 | 67.5 |
| LM Forward | 15M | No | 91.7 | 93.1 | 89.3 | 74.8 | 75.8 | 67.3 |
| Skip-thought | 15M | Yes | 50.7 | 85.4 | 84.9 | 29.6 | 73.8 | 63.5 |
| Skip-thought | 15M | No | 75.2 | 88.1 | 84.9 | 63.5 | 77.4 | 63.7 |
| Autoencoder | 15M | Yes | 77.9 | 82.3 | 81.9 | 66.9 | 68.7 | 62.6 |
| Autoencoder | 15M | No | 61.2 | 80.4 | 82.3 | 56.6 | 69.8 | 62.0 |
| LM (Bidir) | 63M | No | 94.3 | **94.8** | 89.7 | 82.9 | **83.9** | 67.5 |
| LM Forward | 63M | No | 92.1 | 93.3 | 89.4 | 74.9 | 76.2 | 67.6 |
| Skip-thought | 63M | Yes | 69.8 | 90.2 | 86.3 | 55.4 | 78.1 | 64.4 |
| Skip-thought | 63M | No | 77.9 | 89.6 | 86.1 | 64.8 | 78.4 | 64.0 |
| Autoencoder | 63M | Yes | 72.1 | 80.1 | 81.5 | 58.7 | 66.8 | 61.3 |
| Autoencoder | 63M | No | 60.6 | 80.6 | 82.3 | 55.7 | 68.6 | 61.7 |

Table 4: Here we display results for training on 1% of auxiliary task data. Word-conditional most frequent class baselines for this amount of training data are 81.8% for POS tagging and 62.3% for CCG supertagging. For each task, we underline the best performance for each training dataset size and bold the best overall performance.

