# OpenReview forum: "Language Modeling Teaches You More Syntax than Translation Does: Lessons Learned Through Auxiliary Task Analysis"
_ICLR.cc/2019/Conference_

### Official Review · AnonReviewer2 · 2018-11-01

**Rating:** 7
**Confidence:** 4

**Review:**

This is nicely written paper analyzing the effect of various pre-training methods and shows that language models are very effective on sequence tagging tasks (POS, CCG). The experiments are well motivated and well described.

Regarding Table 1: which one of the "LM forward" models was used in the subsequent experiments?

Are the input embeddings for the random init LSTM pre-trained or are they also randomly initialized?

---

> ### Author Response · Authors · 2018-11-18
> **Thanks! Let us know if you have further feedback!**
>
> Thank you!
>
> We used both “LM Forward” models - the larger forward LM we examined on its own and the small forward LM was combined with the small backward LM into the bidirectional LM.
>
> The input embeddings for the randomly initialized LSTMs are also randomly initialized.
>
> Let us know if you have further feedback!

---

### Official Review · AnonReviewer3 · 2018-11-05
**carefully done experiments but is it enough?**

**Rating:** 5
**Confidence:** 4

**Review:**

This paper tests various pretraining objectives (language modeling, machine translation, skip-thought, and autoencoding) on two syntactic tasks: POS tagging and CCG tagging. It finds that language modeling outperforms the other pretraining objectives; additionally, randomly-initializing an encoder achieves decent performance when given a large amount of labeled data for the tagging task. The experiments in this paper are very thorough and explained well. By controlling for pretraining data size, the authors are able to reasonably claim that language modeling is superior to translation as a syntactic transfer learning task. On the other hand, I have some concerns regarding the significance of the paper's contributions, and as such I am borderline on its acceptance.

comments:
- the experiments in the paper feel biased towards language modeling. Language modeling is the only token-level prediction task of the four objectives here, but both of the two downstream tasks are at the token level. It is perhaps unsurprising then that language modeling performs best; perhaps the authors could have considered some sentence-level downstream tasks as well to properly control for this? Or added some more word-level pretraining objectives?

- sort of relatedly, the authors do not provide any explanations as to *why* language modeling is a better pretraining objective than translation. What kinds of examples do the tagging models using LM pretraining get right that the translation models do not? Such an analysis could help provide more concrete insights into what kind of information each objective is encoding.

- the claim that LMs > translation is not a new finding. The authors cite Blevins et al, who find the same result on the task of dependency arc prediction. Similarly, the surprisingly good performance of random encoders was also found in Conneau et al., ACL 2018. As the main contribution of this paper seems to be a more controlled study of Blevins et al on different syntactic tasks, I don't think there is enough here for an ICLR submission.

- what is the effect of the specific dataset and architecture on the results? Here we just look at a couple translation datasets (all news data) and LSTM models. Do things change when we move to transformers or more diverse domains?

---

> ### Author Response · Authors · 2018-11-18
> **Thanks for your feedback!**
>
> - All the tasks use the same training data. The data each model is trained on was pre-processed in the same way, word-level tokenization. All the training objectives we compare all have the same loss: average negative log likelihood of the target sequence. For these reasons, we believe our methods were a fair of all the training tasks we examined. Perhaps I am misunderstanding your comment?
>
> - We agree that why language models are a better pretraining objective than translation is an interesting question. It is in general difficult to answer the question of why representations from one neural model are better than those from another. We did our best to thoroughly compare these training objectives and are interested in any methods / techniques to further address the question of why as there is always room to do more.
>
> - Blevins et al. compare translation models trained on WMT 14 English-German with LMs trained on CoNLL 2012’s training set on dependency arc prediction using the Universal Dependencies dataset. It was unclear from their results alone that if LMs were superior to translation models simply due to differences in the domain and amount of training data for the respective models. We updated how we address this in our literature review section to make this point more clear.
>         The surprisingly good performance of random encoders was found by Conneau et al. ACL 2018 for sentence vector representations using different architectures with pretrained fastText embeddings. We show that randomly initialized LSTM hidden state representations with *randomly* initialized embeddings perform quite well on POS and CCG tagging, which puts the results of Belinkov et al. and Blevins et al. that encoders trained on tasks like MT and language modelling learn syntactic information in a new light, since we find that many of these “learned” syntactic properties can even be learned from random LSTMs. We also show that random LSTMs preserve information about neighboring word identities better than trained LSTMs, which raises new questions about what kind of alternative information is learned by LSTMs that helps them with training task performance.
>
> - We believe that the results for LSTMs trained on WMT data is quite representative of neural NLP models trained on most large-scale datasets. Since we use the same dataset for all tasks, we control for any domain effects. It seems unlikely that using one pretraining task over another will lead to much better or worse domain adaptation from the same domain.

---

> > ### Comment · AnonReviewer3 · 2018-11-25
> > **response**
> >
> > Thanks for the response and the clarification on the random encoder experiments. Just wanted to clarify my point on token vs sentence-level prediction as I think the wording in my initial review was somewhat confusing...
> >
> > "All the tasks use the same training data. The data each model is trained on was pre-processed in the same way, word-level tokenization. All the training objectives we compare all have the same loss: average negative log likelihood of the target sequence. For these reasons, we believe our methods were a fair of all the training tasks we examined. Perhaps I am misunderstanding your comment?"
> >
> > I did not mean that the type of loss was different. I'm referring to language modeling being the only task where a single token-level prediction happens at every position of the input sequence. For all three other tasks (translation, skipthoughts, autoencoding), the entire input is encoded once, and only then is a separate target sequence produced. Your downstream tasks both predict a single label at each position of the input token; as such, language modeling seems better suited as a pretraining task to solve them, as encoders for the other tasks can "ignore" some input tokens without affecting the decoder (something that does not happen in a good language model).  That's why I suggested having some downstream tasks that mirror the setup of the other three pretraining tasks in addition to the two that you already have.

---

> > > ### Author Response · Authors · 2018-11-26
> > > **response**
> > >
> > > Thanks for your clarification. I understand your point now and agree that the per token loss for the language model encoders (in contrast to the encoders for other tasks) could be a reason why LMs perform well on the evaluation tasks. I think the per token loss could also be a reason why LMs are in general able to learn syntax so well.
> > >
> > > I also agree that more evaluation tasks would be better as there is always room to do more. In the LM results section of our most recent revision, I made a note of how our evaluation tasks have per token losses and how LM encoders are the only ones with a per token loss.

---

### Official Review · AnonReviewer1 · 2018-11-06
**Well done with few surprises**

**Rating:** 6
**Confidence:** 4

**Review:**


I have mixed feelings about this paper. On one hand, it’s a thorough and well-written experimental paper, something which is really important but is also clearly underappreciated in the machine learning community. On the other, it was not really obvious to me why some of objectives tested here are interesting: LM objectives like ELMo have seen a lot of uptake in the NLP community (and this is definitely an NLP paper), but most of the others—like skip-thought, MT, and autoencoders—have not. So the basic research question doesn’t seem like an especially burning one. The trends in Fig. 2 show that these alternatives underperform an LM objective, which suggests that the NLP community can keep using that objective without worry—and everything else in the figure seems as we would expect.

In short, I think the paper is a well-done study on a hypothesis of perhaps minor interest. The results are sensible but confirm what we already strongly suspected, and they seem unlikely to strongly influence other research, since they confirm that everyone has been the right thing all along. I’m not entirely sure what I learned from this.

To me, the most interesting experiment is the final one in Section 6. This experiment seems like it could be the germ for a far more interesting paper getting at how these pretraining objectives help with downstream tasks. As it stands, it feels like an interesting nugget tacked on to an otherwise complete (and much less interesting) paper.

Presentational comments:

Fig.1: really nitpicky, but the typography of the POS tags and CCG categories is all wrong. These aren’t mathematical symbols!

Fig 2. Slightly confused why these are broken up into two separate plots.

Fig 4. is hard to read due to the lurid colors and patterns, which require a lot of cross-referencing with the legend. I wonder if this would be better as simply a table. I also found it very confusing at first since the y-axes are out of sync between the two figures—initially it looked as if the legend was overlaid on a set of bars in the left figure that had the same baseline as the right figure.

---

> ### Author Response · Authors · 2018-11-18
> **Thanks for your comments!**
>
> Our contribution is a thorough examination of several different pretraining tasks, controlling for the domain, amount of data, and training procedure. When CoVe was released at NIPS last year, it achieved SoTA numbers on several prominent NLP tasks. Although ELMo compares to CoVe in their paper and outperforms CoVe, since CoVe was trained on WMT English-German and ELMo was trained on the One Billion Word Benchmark, it was unclear if the performance gain of ELMo was primarily due to the increased amount of training data. Moreover, without a direct comparison we can’t even be sure that language modeling is better because ELMo could have just been more carefully tuned. Our finding that language modeling, an unsupervised task, outperformed translation models trained on the *same* data is still surprising because the translation models are given the source sentence in a different language and thus have strictly more information than language models. We also agree that the results of our analysis of the randomly-initialized encoder in Section 6 are surprising, and could form the basis for a larger study.
>
> Fig 1: You are right. We just fixed the typography in our most recent revision.
>
> Fig 2: The upper plot is for POS tagging and the bottom for CCG supertagging. Each of those plots are then split into three columns corresponding to different amounts of classifier training data. Each column has two plots because when we tried plotting all ten lines into one figure it was difficult to read, so we split up the models into two groups: models with attention (plus BiLMs) and models without attention (plus forward LM). We welcome further suggestions for improving our presentation.
>
> Fig 4: We included the patterns and bright colors in order to make it easier for the visually impaired to read.

---

> > ### Comment · AnonReviewer1 · 2018-11-19
> > **Thanks for your response**
> >
> > I agree, it's definitely interesting that CoVe doesn't help, since it has access to strictly more information in your controlled comparison. But suppose you had found the opposite: that training with CoVe worked better than ELMo. Would readers of this paper already using ELMo (or one of its variants) go out and replace it with CoVe? Of course they wouldn't, because they can train ELMo using as much data as they want. With CoVe they would need parallel data, which is much scarcer. In other words: if the results of the experiment had been different, they would likely have the same effect on actual use of these methods, i.e. very little effect at all.
> >
> > Again, this is just a comment on the impact of the paper. I think the science is sound, and I agree it's interesting. Were I chair (I'm not), my decision on this paper would depend on how many other potentially high-impact papers were on offer, and whether there was space left for this one. (I personally favor higher accept rates for precisely this reason. But again it's not up to me.)
> >
> > Re: Fig. 4. It's great that this is intended to be readable to the visually impaired, but I think it fails as an effective figure in general. The main problem is that it requires a lot of cognitive effort to decode what each of the bars means, particularly for (b), since the reader has to cross-reference with the legend on (a). This means scanning back and forth, doing a lot of visual search, and keeping several bits of information in memory at once. Those are exactly the kind of things you want to minimize when presenting information visually.

---

> > > ### Author Response · Authors · 2018-11-23
> > > **Replying to your response**
> > >
> > > Thanks for your response! We understand your viewpoint and also agree that accepting high-impact papers is important. Our paper does have some consequences for decisions about whether it makes sense to try to gather more labeled data in service of pretraining, as it is possible to collect much larger MT datasets than we currently have.
> > >
> > > Also, your argument that language modeling is obviously better since it’s easier to train on more data isn’t so obvious, since large unsupervised datasets alone aren’t always better than smaller amounts of finely labelled data. For example, for the closely related area of learning sentence-vector representations, its been found that that training representations on smaller amounts of supervised data, specifically natural language inference data with only 1 million sentence pairs, is better than using more unlabeled data, specifically training skipthought on the Toronto Books Corpus with 74 million sentences (see Conneau et al. 2017, https://arxiv.org/abs/1705.02364). Similarly, it’s been found that providing explanations rather than simply labels for text classifiers can speed up training time by 5-100x (Hancock et al. 2018, https://arxiv.org/abs/1805.03818). Our paper shows that supervised data in the form of translation pairs is not an efficient way to improve sentence encoder representations.

---

### Meta-Review · Area_Chair1 · 2018-12-14
**Solid study but findings are not very surprising and not very novel**

**Confidence:** 4
**Recommendation:** Reject

**Metareview:**

Strengths:

-- Solid experiments
-- The paper is well written

Weaknesses:

-- The findings are not entirely novel and not so surprising, previous papers (e.g., Brevlins et al (ACL 2018)) have already
suggested that LM objectives are preferable and also using LM objective for pretraining is already the  standard practice (see details in R1 and R3).

There is a consensus between the two reviewers who provided detailed comments and engaged in discussion with the authors.